# A Novel Targeted RIG-I Receptor 5′Triphosphate Double Strain RNA-Based Adjuvant Significantly Improves the Immunogenicity of the SARS-CoV-2 Delta-Omicron Chimeric RBD-Dimer Recombinant Protein Vaccine

**DOI:** 10.3390/v15051099

**Published:** 2023-04-29

**Authors:** Yu Bai, Chaoqiang An, Xuanxuan Zhang, Kelei Li, Feiran Cheng, Bopei Cui, Ziyang Song, Dong Liu, Jialu Zhang, Qian He, Jianyang Liu, Qunying Mao, Zhenglun Liang

**Affiliations:** 1Division of Hepatitis and Enterovirus Vaccines, Institute of Biological Products, National Institutes for Food and Drug Control, NHC Key Laboratory of Research on Quality and Standardization of Biotech Products, NMPA Key Laboratory for Quality Research and Evaluation of Biological Products, Beijing 102600, China; blbaiyu@163.com (Y.B.); zhangxx_01@163.com (X.Z.); 18249518559@163.com (F.C.); cbp31@139.com (B.C.); ld_nifdc@163.com (D.L.); zhang1999jialu@163.com (J.Z.); hq5740@126.com (Q.H.); liujianyang123@126.com (J.L.); 2Shanghai JunTuo Biotechnology Co., Ltd., Shanghai 201210, China; acq0212@gmail.com; 3Beijing Minhai Biotechnology Co., Ltd., Beijning 102600, China; likelei@biominhai.com

**Keywords:** retinoic acid-inducible gene-I (RIG-I) receptor, vaccine adjuvant, immunogenicity, recombinant protein vaccine

## Abstract

The rapid mutation and spread of SARS-CoV-2 variants recently, especially through the emerging variants Omicron BA5, BF7, XBB and BQ1, necessitate the development of universal vaccines to provide broad spectrum protection against variants. For the SARS-CoV-2 universal recombinant protein vaccines, an effective approach is necessary to design broad-spectrum antigens and combine them with novel adjuvants that can induce high immunogenicity. In this study, we designed a novel targeted retinoic acid-inducible gene-I (RIG-I) receptor 5′triphosphate double strain RNA (5′PPP dsRNA)-based vaccine adjuvant (named AT149) and combined it with the SARS-CoV-2 Delta and Omicron chimeric RBD-dimer recombinant protein (D-O RBD) to immunize mice. The results showed that AT149 activated the P65 NF-κB signaling pathway, which subsequently activated the interferon signal pathway by targeting the RIG-I receptor. The D-O RBD + AT149 and D-O RBD + aluminum hydroxide adjuvant (Al) + AT149 groups showed elevated levels of neutralizing antibodies against the authentic Delta variant, and Omicron subvariants, BA1, BA5, and BF7, pseudovirus BQ1.1, and XBB compared with D-O RBD + Al and D-O RBD + Al + CpG7909/Poly (I:C) groups at 14 d after the second immunization, respectively. In addition, D-O RBD + AT149 and D-O RBD + Al + AT149 groups presented higher levels of the T-cell-secreted IFN-γ immune response. Overall, we designed a novel targeted RIG-I receptor 5′PPP dsRNA-based vaccine adjuvant to significantly improve the immunogenicity and broad spectrum of the SARS-CoV-2 recombinant protein vaccine.

## 1. Introduction

Severe acute respiratory syndrome coronavirus 2 (SARS-CoV-2) variants carry mutations in the receptor-binding domain (RBD)-containing spike protein, which can reduce vaccine effectiveness at varying degrees [1,2]. Therefore, there is an urgent need to develop protective vaccines that can induce robust, durable, and broad cross-protective immune responses against multiple variants [3]. One approach to developing a broad-spectrum recombinant protein vaccine is to design broad-spectrum antigens and combine them with novel adjuvants resulting in higher immunogenicity and broader cross-protective immunity [4]. The ligands of pattern recognition receptors have been widely studied as vaccine adjuvants, such as CpG and Poly (I:C), in clinical trials. Both CpG and Poly (I:C) are nucleic acid adjuvants and are used in the development of COVID-19 recombinant protein vaccines [4,5,6,7]. The retinoic acid-inducible gene-I (RIG-I) receptor is widely expressed in immune and non-immune cells and is activated by short 5′triphosphate double strain RNA (5′PPP dsRNA). This leads to the activation of the innate immune system to produce pro-inflammatory cytokines and induce B cell activation, suggesting that RIG-I agonists have potential vaccine adjuvants [8,9]. In our previous study, we found that the RNA sequence GUAU could effectively initiate the RIG-I receptor-induced signaling pathway (unpublished observation). Thus, we designed a novel targeted RIG-I receptor 5′PPP dsRNA, which could improve the immunogenicity of the SARS-CoV-2 Delta-Omicron Chimeric RBD-dimer recombinant protein vaccine.

## 2. Materials and Methods

### 2.1. Biosafety and Ethics Statement

All experiments involving live SARS-CoV-2 were performed in a biosafety level 3 laboratory (BSL-3) of Sinovac Life Sciences Co., Ltd., Beijing, China. All research staff received training on animal experiment operation, care, and handling. The animal study was approved by the Institutional Animal Care and Use Committee of the National Institutes for Food and Drug Control [Approval number: NIFDC-AW-2022(B)056].

### 2.2. 5′PPP dsRNA Preparation

In our previous study, we found that the RNA sequence GUAU could effectively initiate the RIG-I receptor-induced signaling pathway (unpublished observation). Thus, we designed a novel targeted RIG-I receptor 5′PPP dsRNA that contained rich GUAU RNA sequences.

5′PPP dsRNA was produced via in vitro transcription (IVT) by T7 polymerase. The sequence of the dsRNA, called AT149, used in this study was as follows:

5′GAUAGUAUGUAUUAGUAUGUAUGUAUAUUGUAUGUAUGUAUGUAUUAGUAUGUAUGUAUUAGUAUGUAUUAACGGUUAGUGUGUGUUAGUGUGUGUGUGUUAGUGUGUGUGUGUGUGUAAUGUGUGUGUGUGUUAGUGUGUGUUAAUC-3′.

In this study, RNA was prepared by T7 polymerase in vitro transcription, during which three phosphate groups were automatically generated at the 5′ ends.

The molecular length of AT149 was 149nt. The RNA sequence showed a double-stranded hairpin structure predicted by the online biology software RNAfold Web Server (http://rna.tbi.univie.ac.at/cgi-bin/RNAWebSuite/RNAfold.cgi, accessed on 11 April 2023).

The sequences were fused to the PUC57 plasmid with a T7 promoter, and the endonuclease BsmBI-v2 (New England Biolabs, Ipswich, MA, USA) was performed by GENEWIZ (Suzhou, China). IVT was performed for 6 h at 37 °C to produce the required RNA sequences using the T7 RiboMaxTM Express Large-Scale RNA Production System, and RNA was purified according to the manufacturer’s instructions (P1320, Promega, Madison, WI, USA).

### 2.3. Capillary Electrophoresis

In the present study, capillary electrophoresis (Qsep 100, Bioptic lnc., Taiwan, China) was used to test the purity of AT149. The sample was heated at 70 °C for 10 min and then quickly placed on the ice for 3 min. The sample was tested according to the instruction manual of the instrument.

### 2.4. Cell Culture

HEK-Lucia™ Null cells with RIG-I overexpression (OE), RIG-I knockout (KO), and RIG-I Null purchased from InvivoGene (San Diego, CA, USA) are HEK293-derived cells that stably express the Lucia luciferase secreted reporter gene. This reporter gene is under the control of the interferon (IFN)-inducible promoter which is comprised the IFN-stimulated genes (ISG) 54 promoter enhanced by multimeric IFN-stimulated response elements (ISRE). Detailed information is recorded at: https://www.invivogen.com/hek-lucia-null, accessed on 24 April 2023. All cell lines were cultured with DMEM (Gibco, Carlsbad, CA, USA) with 10% Fetal Bovine Serum (10099141, Gibco, Australia), 1% penicillin and streptomycin at 37 °C and 5% CO_2_.

### 2.5. Transfections

At 80% cell confluency, 5 µg AT149 was added to each well for transfecting using the ER4000 liposome transfection reagent according to the manufacturer’s instructions (4000-4, Engreen Biosystem, Beijing, China). Samples were collected at the relevant time points for subsequent experiments.

### 2.6. Luciferase Reporter Assay

Cell culture supernatants were collected at 24 and 48 h post-transfection for the luciferase reporter assay; a 20 µL supernatant and 50 µL fluorogenic substrate (QUANTI-Luc, rep-qlcg1, InvivoGen, San Diego, CA, USA) were mixed, and then luminescence was detected quickly.

### 2.7. Western Blot (WB)

WB was conducted according to the previously established method [10,11]. In this study, gray values for *β*-actin, phosphorylated P65 (*p*-P65) and total P65 were measured using Image J software and were used to calculate the fold increase in *p*-P65.

### 2.8. Vaccine Preparation

SARS-CoV-2 Delta and Omicron BA1 RBD-dimer proteins (D-O RBD) were donated by Anhui Zhifei (Longcom Biopharmaceutical Co., Ltd., Hefei, China). D-O RBD includes one Delta RBD (S protein residues 319–537) and one Omicron BA.1 RBD (S protein residues 316–534) connected in tandem repeats [12]. The doses of AT149 used were 5, 10, 15, and 20 µg. The dose of D-O RBD (one-fifth of the human dose) was 10 µg. The doses of Al, CpG 7909, and poly(I:C) were 100, 50, and 50 µg, respectively, based on the published studies [11,13]. AT149 was encapsulated in lipid nanoparticles (LNP) donated by Beijing Minhai Biotechnology Co., Ltd. (Beijing, China). Because CpG 7909 and poly(I:C) can enter cells through endocytosis, they do not need to be encapsulated in LNP. Adjuvants and D-O RBD were thoroughly mixed and emulsified to prepare the vaccines.

### 2.9. Animal Studies

For Study 1, 30 BALB/c mice (6–8 weeks old, 18–20 g) were randomly and equally assigned to six groups, as detailed in Table 1. For Study 2, 60 BALB/c mice (6–8 weeks, 18–20 g) were randomly assigned to ten groups, as detailed in Table 2. An overview of the experimental procedure for both studies is shown in Figure 1.

### 2.10. Enzyme-Linked Immunosorbent Assay (ELISA)

ELISA was used to measure IgG titers in the serum. The experimental process was based on the previously established method [10,14].

### 2.11. Serum Neutralization Assay

A neutralization assay for authentic viruses was performed to measure neutralizing antibody (NAb) titers, as described previously [14]. A neutralization assay for pseudoviruses was performed to measure NAb titers, which were displayed as the 50% inhibitory dilution (EC_50_) of the serum, as described previously [15].

### 2.12. IFN-γ ELISPOT Assay

IFN-γ secreted by splenocytes was detected using the Mouse IFN-γ ELISPOT kit (BD, 552569, San Diego, CA, USA) according to the manufacturer’s instructions [11,14].

### 2.13. Statistical Significance

Statistical analyses were performed using GraphPad Prism 8.0. Statistical analyses were performed between the two groups using Student’s *t*-test and between at least three groups using Tukey’s multiple comparison test. The statistical value of *p* < 0.05 indicated a statistically significant difference.

## 3. Results and Discussion

### 3.1. AT149 Activated Innate Immunity by Targeting the RIG-I Signaling Pathway

In the present study, we designed a novel targeted RIG-I receptor 5′PPP dsRNA named AT149 with a hairpin structure predicted by RNAfold Web Server software (Figure 2A). Gel electrophoresis revealed that the molecular length of AT149 was between 100 and 200 nt (Figure 2B). The capillary electrophoresis results showed the prepared AT149 eluted as a single peak with a purity of 86.7%, indicating that the product was simple with high purity. The pink peak was a 20 bp marker that came with the instrument Qsep 100. The blue peak was the sample peak (Figure 2C). We found that AT149 could dramatically activate the interferon signaling pathway in HEK293 with RIG-I overexpression rather than RIG-I knockout (KO) and RIG-I null. The P65 NF-κB (P65) signal pathway is the classical pro-inflammatory signaling pathway, which plays a critical role in innate immunity. Additionally, phosphorylated P65 (*p*-P65) is the active form of P65 NF-κB. The increase in *p*-P65 meant that this pro-inflammatory signaling pathway was activated. In the present study, AT149 transfection could increase RIG-I and the *p*-P65 expression, suggesting that AT149 5′PPP dsRNA could induce RIG-I increasing to activate the following P65 NF-κB pro-inflammatory signaling pathway. These results confirm that AT149 can activate innate immunity by targeting the RIG-I receptor-induced signaling pathway (Figure 2D,E).

### 3.2. AT149 +/Al Adjuvant Significantly Improved the Immunogenicity of the SARS-CoV-2 Delta-Omicron Chimeric RBD-Dimer Recombinant Protein Vaccine

Subsequently, the AT149 adjuvanted Delta-Omicron BA1 chimeric RBD-dimer recombinant protein was used to immunize mice two times with one-fifth of the human dosage (Figure 1). The genetic unrooted phylogenic tree of SARS-CoV-2 strains was constructed by Nextstrain according to the sequences of 2802 SARS-CoV-2 spike glycoproteins from GISAID collected between December 2019 and April 2023. The result indicated that Omicron variants showed the highest antigenic distinction to the ancestral strain, Delta, Beta and other variants (Figure 3A). At 14 d post-first immunization, Delta or Omicron BA1 protein-specific IgG titers in D-O RBD + AT149 (5–20 µg) groups were substantially higher than that of the D-O RBD + Al group in a dose-dependent manner. Combination adjuvants more effectively improved immunogenicity and are gradually being used in recombinant protein vaccines, such as MF59, Matrix-MTM and AS01. Poly (I:C) and CpG are typical nucleic acid adjuvants, which are widely studied and, in the present study, presented good adjuvant effects. In clinical trials, Al + CpG7909 has been used as a compound adjuvant in SARS-CoV-2 recombinant protein vaccines [10]. In the present study, Al and AT149 were combined. The results showed higher Delta and Omicron BA1 specific IgG titers for D-O RBD with Al + AT149 (5–20 μg) compared to D-O RBD + Al + CpG7909/Poly (I:C) groups (Figure 3B).

NAb titers against authentic viruses are crucial indicators of the immunogenicity of COVID-19 vaccines [16]. At 14 d post-second immunization, NAb titers against authentic Delta and Omicron BA1 of D-O RBD + AT149 (5–20 µg) were significantly increased from 13- to 59-fold and 16- to 38-fold compared to that of D-O RBD + Al in a dose-dependent manner, respectively. Surprisingly, cross-neutralizing antibody titers against Omicron BA5 and BF7 were also increased from 7- to 21-fold in the D-O RBD + AT149 (5–20 μg) groups and were 21- to 31-fold higher than in the D-O RBD + Al group (Figure 3C). The latest variants, Omicron BQ1 and XBB present stronger transmissibility and immune evasion ability than other variants and are gradually circulating worldwide [1]. NAb titers against pseudovirus BQ1.1 and XBB of D-O RBD + AT149 (15 µg) groups were also increased 2-fold compared to that of the D-O RBD + Al group (Figure 3D). Additionally, NAb titers against authentic Delta and Omicron BA1 of D-O RBD + Al + AT149 (5–20 µg) were also substantially increased from 53- to 86-fold and 37- to 84-fold compared to that of D-O RBD + Al, respectively, and were also significantly higher than D-O RBD + Al + CpG 7909/Poly (I:C) groups (Figure 3E). Meanwhile, D-O RBD + Al + AT149 (5–20 µg) increased cross-neutralizing antibody titers against Omicron BA5 BF7 compared to D-O RBD + Al + CpG 7909/Poly (I:C) (Figure 3E). NAb titers against pseudovirus BQ1.1 and XBB of D-O RBD + Al + AT149 (10–15 µg) groups were also increased in mice (Figure 3F). Omicron variants showed the highest antigenic distinction to the ancestral strain, Delta, Beta, and other variants. Additionally, a large variation among Omicron BA1, BA5, BF7, BQ1.1 and XBB could also be observed. Published studies found that the Delta variant presented the strongest pathogenicity, and the Delta variant RBD as an antigen could induce the more broad-spectrum Nab compared with other virus strains [13,17,18]. Thus, we selected the Delta and Omicron BA1 RBD dimer recombinant protein as an antigen in this study. The results of this study indicate that AT149 and AT149 + Al can robustly improve the humoral immune responses of the SARS-CoV-2 Delta-Omicron chimeric RBD-dimer recombinant protein vaccine in mice, and the vaccine, when compatible with the AT149/+ Al novel adjuvant, provided more effective cross-neutralization against Omicron BA5, BF7, BQ1.1, and XBB.

The Th1 immune response can activate the CD8^+^ T immune response, which plays a critical role in destroying viruses through cytotoxic T lymphocytes [2,3]. The T-cell-secreted IFN-γ immune response is representative of the Th1 immune response. In the present study, D-O RBD + AT149 and D-O RBD + Al + AT149 groups improved the Omicron BA1 RBD-specific T-cell-secreted IFN-γ immune response when compared with D-O RBD + Al and D-O RBD + Al + CpG 7909/Poly (I:C) groups, respectively (Figure 3G). Indeed, published studies show that one of the concerns with COVID-19 vaccines has been the risk of vaccine-enhanced disease through a Th2 >> Th1 immune response [4,19,20,21]. We will measure the Th2 response in future studies.

One reason for the “self-adjuvant” effect of mRNA vaccines is that LNP can activate the innate immune system post-vaccination [16]. We also observed that the LNP used in our study could perform the adjuvant function for the recombinant protein vaccine to induce a higher level of NAb compared to the Al adjuvant, suggesting not only that LNP has the ability to act as a delivery system for mRNA vaccines but also as a potential adjuvant for the recombinant protein vaccine (Figure 3E). In our previous study, we found that the RIG-I receptor-induced signaling pathway could produce a lot of IFN to exercise a broad-spectrum antiviral function when post-activated by 5′PPP dsRNA [11]. Thus, AT149 might also perform this biological function, which will be studied in the future.

## 4. Conclusions

The novel targeted RIG-I receptor 5′PPP dsRNA-based vaccine adjuvant AT149 designed in this study can significantly improve the immunogenicity and broaden the spectrum of the SARS-CoV-2 Delta-Omicron chimeric RBD-dimer recombinant protein vaccine.

## Figures and Tables

**Figure 1 viruses-15-01099-f001:**
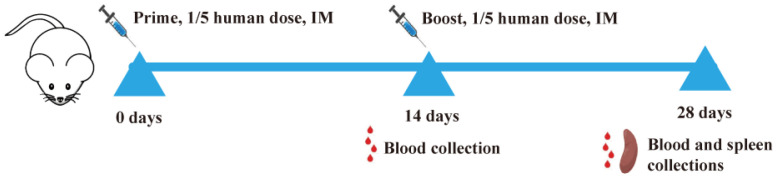
A high-level overview of the animal study design.

**Figure 2 viruses-15-01099-f002:**
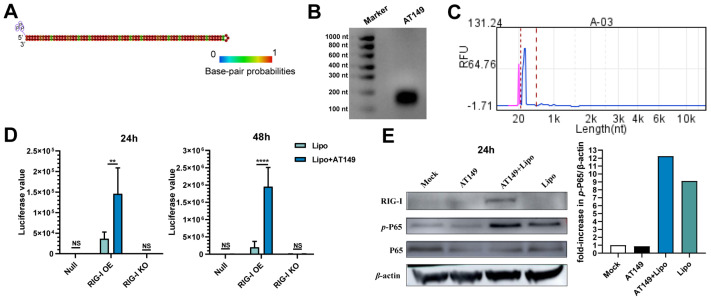
Innate immunity response to AT149 via RIG-I receptor-induced signaling pathway. (**A**) Hairpin structure of AT149 was predicted by RNAfold Web Server software. (**B**) Denatured agarose gel electrophoresis of AT149. (**C**) The result of capillary electrophoresis. The pink peak is a 20 bp marker that comes with the instrument Qsep 100. The blue peak is the sample peak. (**D**) Relative fluorescence intensity in vitro at 24 and 48 h after transfection. (**E**) WB results in A549 at 24 h after transfection and a similar result was obtained by the two independent experiments. **, *p* < 0.01. ****, *p* < 0.0001. NS, non-significance. Lipo, liposome (ER4000 used in this study).

**Figure 3 viruses-15-01099-f003:**
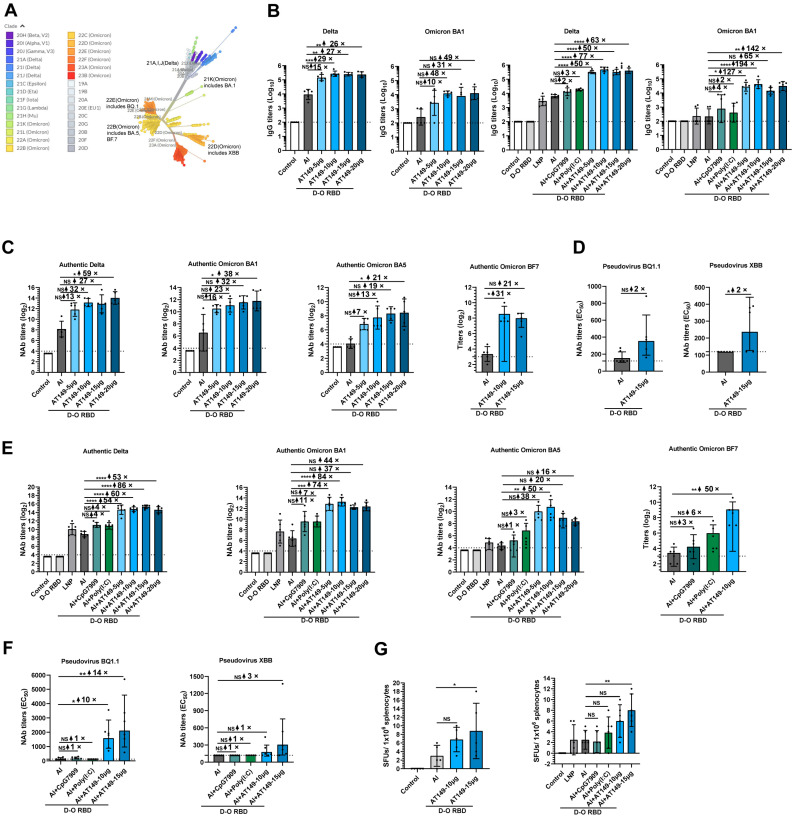
Significantly improved immunogenicity and cross-neutralization of the SARS-CoV-2 Delta-Omicron chimeric RBD-dimer recombinant protein vaccine in AT149 and AT149 + Al treatment. (**A**) Genetic relationships of global SARS-CoV-2 strains. The unrooted phylogenic tree was constructed by Nextstrain according to the sequences of 2802 SARS-CoV-2 spike glycoproteins from GISAID collected between December 2019 and April 2023 (https://nextstrain.org/ncov/gisaid/global/6m?l=unrooted&m=div, accessed on 24 April 2023). The tree was colored by clade and branch length and represented the divergence of each clade. The Omicron variants show the highest antigenic distinction to the ancestral strain. (**B**) Delta and Omicron BA1 RBD-specific IgG titers in mice at 14 d post-first immunization. (**C**) NAb titers against authentic Delta, Omicron BA1, BA5 and BF7 in Control, D-O RBD + Al, and D-O RBD + AT149 (5–20 µg) groups at 14 d post-second immunization. (**D**) NAb titers against pseudovirus BQ1.1 and XBB of D-O RBD + Al and D-O RBD + AT149 (15 µg) groups at 14 d post-second immunization. (**E**) NAb titers against authentic Delta, Omicron BA1, BA5 and BF7 in Control, D-O RBD, D-O RBD + LNP, D-O RBD + Al, D-O RBD + Al + CpG7909, D-O RBD + Al + Poly (I:C), and D-O RBD + Al + AT149 (5–20 µg) groups at 14 d post-second immunization. (**F**) NAb titers against pseudovirus BQ1.1 and XBB of D-O RBD + Al, D-O RBD + Al + CpG7909, D-O RBD + Al + Poly (I:C), and D-O RBD + Al + AT149 (10–15 µg) groups at 14 d post-second immunization. (**G**) Omicron BA1 RBD-specific T cells secreted IFN-γ in mice 14 d post-second immunization. NS, non-significance. * *p* < 0.05. ** *p* < 0.01. *** *p* < 0.001. **** *p* < 0.0001. ↑, increase. *n* = 5–6 for each group.

**Table 1 viruses-15-01099-t001:** Detailed information of groups in Study 1.

Group	Vaccination	
	Day 0	Day 14
Control	Normal saline	Normal saline
D-O RBD + Al	D-O RBD + Al	D-O RBD + Al
D-O RBD + AT149 (5 µg)	D-O RBD + AT149 (5 µg)	D-O RBD + AT149 (5 µg)
D-O RBD + AT149 (10 µg)	D-O RBD + AT149 (10 µg)	D-O RBD + AT149 (10 µg)
D-O RBD + AT149 (15 µg)	D-O RBD + AT149 (15 µg)	D-O RBD + AT149 (15 µg)
D-O RBD + AT149 (20 µg)	D-O RBD + AT149 (20 µg)	D-O RBD + AT149 (20 µg)

D-O RBD: SARS-CoV-2 Delta and Omicron BA1 RBD-dimer proteins, Al: aluminum hydroxide adjuvant.

**Table 2 viruses-15-01099-t002:** Detailed information of groups in Study 2.

Group	Vaccination	
	Day 0	Day 14
Control	Normal saline	Normal saline
D-O RBD	D-O RBD	D-O RBD
D-O RBD + LNP	D-O RBD + LNP	D-O RBD + LNP
D-O RBD + Al	D-O RBD + Al	D-O RBD + Al
D-O RBD + Al + CpG 7909	D-O RBD + Al + CpG 7909	D-O RBD + Al + CpG 7909
D-O RBD + Al + poly(I:C)	D-O RBD + Al + poly(I:C)	D-O RBD + Al + poly(I:C)
D-O RBD + Al + AT149 (5 µg)	D-O RBD + Al + AT149 (5 µg)	D-O RBD + Al + AT149 (5 µg)
D-O RBD + Al + AT149 (10 µg)	D-O RBD + Al + AT149 (10 µg)	D-O RBD + Al + AT149 (10 µg)
D-O RBD + Al + AT149 (15 µg)	D-O RBD + Al + AT149 (15 µg)	D-O RBD + Al + AT149 (15 µg)
D-O RBD + Al + AT149 (20 µg)	D-O RBD + Al + AT149 (20 µg)	D-O RBD + Al + AT149 (20 µg)

D-O RBD: SARS-CoV-2 Delta and Omicron BA1 RBD-dimer proteins, Al: aluminum hydroxide adjuvant.

## Data Availability

All data were generated or analyzed, and the materials used in this study are included in this published article.

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
