# Peer review of "A Novel Targeted RIG-I Receptor 5′Triphosphate Double Strain RNA-Based Adjuvant Significantly Improves the Immunogenicity of the SARS-CoV-2 Delta-Omicron Chimeric RBD-Dimer Recombinant Protein Vaccine"

_viruses, 2023, doi:10.3390/v15051099_

Round 1
Reviewer 1 Report
In this study, researchers designed a novel targeted RIG-I receptor dsRNA-based adjuvant, which not only improved the immunogenicity against Delta and Omicron BA1 variants, but also enhanced the cross-neutralization ability against Omicron subvariants, BA5, BF7, BQ1.1, and XBB circulating recently in the SARS-CoV-2 Delta-Omicron BA1 chimeric RBD-dimer recombinant protein vaccine. On the one hand, these results indicated that the novel targeted RIG-I receptor dsRNA-based adjuvant induced high immunogenicity and can also improve the broad spectrum of the SARS-CoV-2 recombinant protein vaccine, with promising implications in the future. On the other hand, these findings provide bases for the research and development of novel vaccine adjuvants. This paper is suitable for viruses. Thus, I advise that this paper should be accepted and published in viruses.
There are some minor revisions need authors to be done. The details are in the following.
(1) In the figure 1A, it showed that the dsRNA AT149 present a hairpin structure predicted by RNA fold Web Server software. In your study, you papered dsRNA by T7 polymerase. The RNA produced by T7 polymerase has three phosphate groups. And three phosphate groups play an important role in activating RIG-I signal pathway. Thus, please draw the three phosphate groups in your figure.
(2) In this study, the researchers used Poly(I:C) and CpG7909 as comparison groups, please try to explain the reason.
(3) As we all know that when RIG-I signal pathway is activated, IFN will produce subsequently. Please try to discuss whether dsRNA can be used as anti-virus medicine or others. That will improve the valuable of your design dsRNA.
(4) Please provide more data about dsRNA characterization, such as purity, the exact size in adjuvant.
(5) It will be better if put LNP+AT149 group in study 2.
Minor improvement in English will be required.
Author Response
Responses to reviewer 1’ comments for the manuscript (NO. viruses-2366845)
Dear Editor/ reviewer,
We are very grateful to editor and reviewer for the interest in our work and those valuable advices and comments. We have adopted the suggestion and revised them accordingly. The revised parts were marked in green in the revised manuscript.
Reviewer Comments
(1)In the figure 1A, it showed that the dsRNA AT149 present a hairpin structure predicted by RNA fold Web Server software. In your study, you papered dsRNA by T7 polymerase. The RNA produced by T7 polymerase has three phosphate groups. And three phosphate groups play an important role in activating RIG-I signal pathway. Thus, please draw the three phosphate groups in your figure.
Response: Thanks for this helpful suggestion. In the revised manuscript, we have drawn the three phosphate groups at the 5’end of RNA in figure 1 A.
(2)In this study, the researchers used Poly(I:C) and CpG7909 as comparison groups, please try to explain the reason.
Response: Thanks for this helpful comment. We added the description in the revised manuscript. “Poly (I:C) and CpG are the typical nucleic acid adjuvants, which are widely studied and presented good adjuvant effects now. In clinical trials, Al + CpG7909 has been used as a compound adjuvant in SARS-CoV-2 recombinant protein vaccines” (Line 200-203, Page 5)
(3) As we all know that when RIG-I signal pathway is activated, IFN will produce subsequently. Please try to discuss whether dsRNA can be used as anti-virus medicine or others. That will improve the valuable of your design dsRNA.
Response: Thanks for this valuable suggestion. We added the discussion in the revised manuscript. “In our previous study, we found that RIG-I induced signal pathway could produce a good deal of IFN post activated by 5’PPP dsRNA, and exercise broad-spectrum antiviral function. Thus, AT149 might also perform this biological function. That will be studied in the future.” (Line 249-252, Page 6)
(4)Please provide more data about dsRNA characterization, such as purity, the exact size in adjuvant.
Response: Thanks for this valuable comment. We added the data about dsRNA purity by capillary electrophoresis in the revised manuscript. The capillary electrophoresis results showed that the prepared AT149 was a single peak and the purity was 86.7%, indicating that the product was simple with high purity. The pink peak is a 20bp marker that comes with the instrument Qsep 100. The blue peak is the sample peak. (Line 165-168, Page 4). And the experiment procedure of capillary electrophoresis was added into the Materials and Methods. (Line 92-95, Page 2).
(5) It will be better if put LNP+AT149 group in study 2.
Response: Thanks for this valuable suggestion. LNP + AT149 could activate the innate immunity rather than the specificity antibodies. We will add this group in the future research.
We really appreciate for the earnest work of Editors and Reviewers, and hope that the corrections will meet with approval. Once again, thank you very much for your comments and suggestions. We look forward to your information about my revised manuscript and thank you for your good comments.
Best regards!
Yours sincerely,
Dr. Zhenglun Liang
National Institutes for Food and Drug Control. Huatuo Road No.31, Daxing District, Beijing, China. 102629
Email: [email protected]

Reviewer 2 Report
The rapid evolution of SARS-CoV-2 viruses prompted the development of universal vaccines, along with adjuvants capable of inducing stronger immune responses. The investigators designed a novel vaccine with retinoic 21 acid-inducible gene-I (RIG-I) receptor adjuvant (named AT149) in order to improve immunogenicity of SARS-CoV-2 Delta and omicron chimeric RBD-dimer recombinant protein (D-O RBD). They demonstrated that AT149 can activate the P65 NF-κB signaling pathway involving in the interferon signal pathway by targeting RIG-I receptor. The D-O RBD + AT149 and D-O RBD + aluminum hydroxide adjuvant (Al) + AT149 groups increased the levels of neutralizing antibodies against Delta variant, and Omicron subvariants, BA1, BA5, and BF7, pseudovirus BQ1.1, and XBB compared with D-O RBD + Al and D-O RBD + Al + CpG7909/ Poly (I:C) groups respectively. In addition, D-O RBD + AT149 and D-O RBD + Al + AT149 groups induced higher levels of IFN-γ immune response.
This is an interesting piece of study which presents appreciable novelty. Just a few comments for the authors to consider in the revised manuscript
1. Title “visible” could be changed to “ significant”
2. Abstratc: line 18, change “urge” to “ necessitate”… there are many syntax issues throughout the draft that could be edited.
3. Line 64; “Data not published now” change to “unpublished observation”
4. Fig.1D , the WB could be considered to be scanned for density with numerical values presented as graphs,
5. Fig 2 depicts the main data which is clear, preferably the authors may wish to describe how different the variants are in terms of phylogenetic tree, and more discussion should be considered.
6. The manuscript could benefit from being edited by a native English speaker to correct syntax errors in the manuscript.
The rapid evolution of SARS-CoV-2 viruses prompted the development of universal vaccines, along with adjuvants capable of inducing stronger immune responses. The investigators designed a novel vaccine with retinoic 21 acid-inducible gene-I (RIG-I) receptor adjuvant (named AT149) in order to improve immunogenicity of SARS-CoV-2 Delta and omicron chimeric RBD-dimer recombinant protein (D-O RBD). They demonstrated that AT149 can activate the P65 NF-κB signaling pathway involving in the interferon signal pathway by targeting RIG-I receptor. The D-O RBD + AT149 and D-O RBD + aluminum hydroxide adjuvant (Al) + AT149 groups increased the levels of neutralizing antibodies against Delta variant, and Omicron subvariants, BA1, BA5, and BF7, pseudovirus BQ1.1, and XBB compared with D-O RBD + Al and D-O RBD + Al + CpG7909/ Poly (I:C) groups respectively. In addition, D-O RBD + AT149 and D-O RBD + Al + AT149 groups induced higher levels of IFN-γ immune response.
This is an interesting piece of study which presents appreciable novelty. Just a few comments for the authors to consider in the revised manuscript
1. Title “visible” could be changed to “ significant”
2. Abstratc: line 18, change “urge” to “ necessitate”… there are many syntax issues throughout the draft that could be edited.
3. Line 64; “Data not published now” change to “unpublished observation”
4. Fig.1D , the WB could be considered to be scanned for density with numerical values presented as graphs,
5. Fig 2 depicts the main data which is clear, preferably the authors may wish to describe how different the variants are in terms of phylogenetic tree, and more discussion should be considered.
6. The manuscript could benefit from being edited by a native English speaker to correct syntax errors in the manuscript.
Author Response
Responses to reviewer 2’ comments for the manuscript (NO. viruses-2366845)
Dear Editor/ reviewer,
We are very grateful to editor and reviewer for the interest in our work and those valuable advices and comments. We have adopted the suggestion and revised them accordingly. The revised parts were marked in green in the revised manuscript.
Reviewer Comments
- Title “visible” could be changed to “significant”
Response: Thanks for this suggestion. We have changed it in the revised manuscript.
- Abstract: line 18, change “urge” to “necessitate”. there are many syntax issues throughout the draft that could be edited.
Response: Thanks for this suggestion. We have changed it in the revised manuscript. And the whole manuscript was edited by a native English speaker to correct syntax errors.
- Line 64; “Data not published now” change to “unpublished observation”
Response: Thanks for this suggestion. We have changed it in the revised manuscript.
- 1D, the WB could be considered to be scanned for density with numerical values presented as graphs.
Response: Thanks for this suggestion. The WB was scanned by Image J software for density with numerical values. And we presented it as graphs in Figure 1 E in the revised manuscript.
- Fig 2 depicts the main data which is clear, preferably the authors may wish to describe how different the variants are in terms of phylogenetic tree, and more discussion should be considered.
Response: Thanks for this helpful suggestion. In the revised manuscript, we added the variants are in terms of phylogenetic tree in Figure 2. And we added the discussion to improve the significance of scientific research. “The Omicron variants show the highest antigenic distinction to the ancestral strain, Delta, Beta and other variants. And also, there is large variation among Omicron BA1, BA5, BF7, BQ1.1 and XBB. The published study found that Delta variant presents the stronger pathogenicity. The published study showed that Delta variant RBD as antigen could induce the more broad-spectrum Nab compared with other virus strains. Thus, we selected Delta and Omicron BA1 RBD dimer recombinant protein as antigen in this study.” (Line 225-231, Page 6).
- The manuscript could benefit from being edited by a native English speaker to correct syntax errors in the manuscript.
Response: Thanks for this suggestion. The whole manuscript was edited by a native English speaker to correct syntax errors.
We really appreciate for the earnest work of Editors and Reviewers, and hope that the corrections will meet with approval. Once again, thank you very much for your comments and suggestions. We look forward to your information about my revised manuscript and thank you for your good comments.
Best regards!
Yours sincerely,
Dr. Zhenglun Liang
National Institutes for Food and Drug Control. Huatuo Road No.31, Daxing District, Beijing, China. 102629
Email: [email protected]

Reviewer 3 Report
Review Viruses-2366845, Bai et al
Bai et al. describe a novel dsRNA molecule, AT149, that can activate the RIG-1 receptor, and its effect on the immune responses induced by a recombinant protein COVID-19 vaccine. They show mouse experiments that when combined with AT149, with or without Alum Hydroxide, the protein-based vaccine induces stronger humoral and cellular immune responses, compared to the combination with Alum Hydroxide alone.
Main comment
It is difficult to interpret the importance of the findings. On one hand, it is not clear if AT149 is an optimal dsRNA. What would the effect of a shorter or longer hairpin be? Perhaps these data are available in the unpublished results that the authors refer to. On the other hand, how do the responses compare to currently approved vaccines? Inclusion of a comparator vaccine or a panel of human convalescent sera would greatly facilitate the interpretation of the results.
Further comments
Lines 66-69 - RNA contains U instead of T.
Lines 160-161 – The characterization of AT149 by gel electrophoresis is insufficient. The authors should perform a more in-depth analysis to show which percentage of AT149 is full-length and the distribution of incomplete transcripts that are likely to be present. Could the authors comment on the possibility that AT149 forms full-length dimers as the sequence is almost completely complementary over the full length of the molecule?
Line 163 – The authors should explain the findings of the A549 experiment in more detail. It should be explained what ‘p-P65’ represents and the authors should comment on the apparent upregulation of RIG-1 (upper panel, lane 3). The 21.97- and 11.15-fold fold increase that is indicated is not apparent from the western blot that is being presented. How were the bands quantified to calculate the p-P65/P65 ratio?
Line 173 – The authors use the word ‘visibly’ several times. Since they perform statistical significance testing, the word ‘significantly’ would be better. Check throughout manuscript.
Line 212 – Have the authors also measured any Th2 responses? One of the concerns with the COVID-19 vaccines has been the risk of vaccine-enhanced disease through a Th2 >> Th1 immune response.
Figure 2A depicts a high-level overview of the study design, not a ‘detailed experimental procedure’.
References – check references for consistency. For example, some references contain full journal names and full first names instead of initials.
Suggestions for changes to the text
Lines 42-45 – The authors mention that ‘The accepted approach to develop a broad spectrum recombinant protein vaccine is to design broad-spectrum antigens and combine them with more effective novel adjuvants that can generate higher immunogenicity and broader cross-protective immunity’. However, to increase the breadth of vaccines, most often the number of antigens is increased (e.g., Prevnar and Gardasil). Suggestion to amend text to: ‘One approach to develop a broad-spectrum recombinant protein vaccine is to design broad-spectrum antigens and combine them with adjuvants resulting in higher immunogenicity and broader cross-protective immunity [4].’
Lines 45-46 - CpG or poly(I:C) have been used in clinical trials for many years, so ‘Recently,’ should be deleted. Furthermore, it is not clear why these two adjuvants are mentioned.
Lines 47-48 - Suggestion to change to: ‘The retinoic acid-inducible gene-I (RIG-I) receptor is widely expressed in immune and non-immune cells and is activated by short 5′triphosphate double strain RNA (dsRNA). This will lead to activation of the innate immune system to produce pro-inflammatory cytokines and induce B cell activation, suggesting that RIG-1 agonists have potential as vaccine adjuvants [8, 9].
Line 51 – Please, clarify to what GUAU was compared or delete ‘more’ from the sentence.
Line 52 – Check instruction for authors on referencing unpublished work.
Lines 63-65 – Delete sentences, as they are already mentioned in the introduction.
Lines 66-69 – Amend to: ‘dsRNA was produced via in vitro transcription (IVT). The sequence of the dsRNA, called AT149, used in this study was as follows:
5’GAUUAGUAUGUAUUAGUAUGUAUGUAUAUUGUAUGUAUGUAUGUAUUAGUAUGUAUGUAUUAGUAUGUAUUAACGGUUAGUGUGUGUUAGUGUGUGUGUGUUAGUGUGUGUGUGUGUGUAAUGUGUGUGUGUGUUAGUGUGUGUUAAUC-3’
The authors mention that AT149 ‘was designed based on our previous study, which activated the RIG-I signaling pathway substantially [10].’ However, it is not clear from reference 10 why a GUAU-rich sequence was chosen.
Lines 84-85 – Please, provide exact names of HEK293 reporter cell lines from Invivogen and check the description of the reporter cell lines. Note that the Invivogen website mentions that in HEK-Lucia cell lines, Luciferase expression is under the control of the interferon (IFN)-inducible promoter which is comprised of the IFN-stimulated genes (ISG) 54 promoter (see https://www.invivogen.com/hek-lucia-null).
Line 108 – amend to ‘AT149 cells were was encapsulated….’
Lines 114-125 – The rationale for the experiment should be mentioned in the Results section. Proposed change: ‘For study 1, 30 Balb/c mice (6–8 weeks old, 18–20 g) were randomly and equally assigned to six groups as detailed in Table 1. For Study 2, 60 BALB/c mice (6–8 weeks, 18–20 g) were randomly assigned to ten groups as detailed in Table 2. An overview of the experimental procedure for both studies is shown in Figure 2A.’
Lines 158-160 – Delete ‘and the abundant GUAU sequence to investigate whether it could improve the immunogenicity and the broad spectrum of recombinant protein vaccines’.
Line 173 - Delete ’could’.
Line 175 – Suggestion: ‘Subsequently, AT149 adjuvanted Delta-Omicron BA1 chimeric RBD-dimer recombinant protein (D-O RBD) was used to immunize mice two times with one-fifth of the human dosage (Figure 2 A). ’
Lines 183-186 – Suggestion: ‘In the present study, Al and AT149 were combined. Results showed higher Delta and Omicron BA1 specific IgG titers for D-O RBD with Al + AT149 (5–20 μg) compared to D-O RBD + Al + CpG7909/ Poly (I:C) groups (Figure 2B).’
Line 190: ‘13- to 59-fold’ and ‘16- to 38-fold’.
Lines 191-194 – Suggestion: ‘Surprisingly, cross-neutralizing antibody titers against Omicron BA5 and BF7 were also increased by 7- to 21-fold in the D-O RBD + AT149 (5–20 μg) groups and were 21- to 31- fold higher than in the D-O RBD + Al group (Figure 2C).’
Suggestion to mover comments on transmissibility and immune evasion of variants from lines 191-192 and 195-197 to line 209: ‘….. and provide more effective cross-neutralization against recently emerged, highly transmissible variants Omicron BA5, BF7, BQ1.1, and XBB, which have shown increased immune evasion.’
Line 211 – what does ‘resisting SARS-CoV-2 invasion’ mean?
Figure 2 – A larger figure is needed to be able to read some of the text within the figure. Upon printing, the fold increase that is mentioned in most panels is hardly readable.
Line 226 – see earlier comment on ‘high-level overview’ vs ‘detailed experimental procedure’.
Line 240 – ‘…broaden the spectrum…’
See 'Suggestions for changes to the text' in the comments above
Author Response
Responses to reviewer 3’ comments for the manuscript (NO. viruses-2366845)
Dear Editor/ reviewer,
We are very grateful to editor and reviewer for the interest in our work and those valuable advices and comments. We have adopted the suggestion and revised them accordingly. The revised parts were marked in green in the revised manuscript.
Reviewer Comments
Main comment
It is difficult to interpret the importance of the findings. On one hand, it is not clear if AT149 is an optimal dsRNA. What would the effect of a shorter or longer hairpin be? Perhaps these data are available in the unpublished results that the authors refer to. On the other hand, how do the responses compare to currently approved vaccines? Inclusion of a comparator vaccine or a panel of human convalescent sera would greatly facilitate the interpretation of the results.
Response: Thanks for this valuable comment. In our unpublished results, we have optimized the sequence and length of dsRNA. We found that the dsRNA with abundant GUAU sequence and the length was more than 100nt that could effectively activate RIG-I signal pathway. In fact, we don't have to find the best sequence and length of dsRNA. It is enough that we just have to found the dsRNA could effectively activate the RIG-I signal pathway and it as adjuvant could improve the immunogenicity of vaccine.
In the present study, SARS-CoV-2 Delta and Omicron BA1 RBD-dimer proteins (D-O RBD) used in this study were donated by Anhui Zhifei (Longcom Biopharmaceutical Co., Ltd., Hefei, China). In fact, D-O RBD + Al is Anhui Zhifei’s variant vaccine that is in clinical trials. In this study, our main goal is to research and develop novel adjuvants. Thus, we select two kinds of nucleic acid vaccine adjuvants, CpG7909 and Poly (I: C) as comparable groups. Besides, CpG7909 + Al as adjuvant have been used for SARS-CoV-2 recombinant protein vaccine, which was in clinical trial. In the revised manuscript, we added the description of two kinds of nucleic acid vaccine adjuvants as comparable groups to greatly facilitate the interpretation of the results. “Ligands of pattern recognition receptors are widely studied as vaccine adjuvants, such as CpG and Poly (I:C) in clinical trial. Both CpG and Poly (I:C) are nucleic acid adjuvants and they are used in the development of COVID-19 recombinant protein vaccines” (Line 53-56, Page 2).
Further comments
- Lines 66-69 - RNA contains U instead of T.
Response: Thanks for this comment. We have modified it in the revised manuscript.
- Lines 160-161 – The characterization of AT149 by gel electrophoresis is insufficient. The authors should perform a more in-depth analysis to show which percentage of AT149 is full-length and the distribution of incomplete transcripts that are likely to be present. Could the authors comment on the possibility that AT149 forms full-length dimers as the sequence is almost completely complementary over the full length of the molecule?
Response: Thanks for this valuable comment. The capillary electrophoresis results showed that the prepared AT149 was a single peak and the purity was 86.7%, indicating that the product was simple with high purity. The pink peak is a 20bp marker that comes with the instrument Qsep 100. The blue peak is the sample peak. (Line 165-168, Page 4). We did not observe this phenomenon by gel electrophoresis and capillary electrophoresis. Thus, it might not exist the dimer or the content of dimer is too low to be detected. The reason of this phenomenon might be that the hairpin structure is the most stable in this system. That's something we'll study in the future. Thank you again.
- Line 163 – The authors should explain the findings of the A549 experiment in more detail. It should be explained what ‘p-P65’ represents and the authors should comment on the apparent upregulation of RIG-1 (upper panel, lane 3). The 21.97- and 11.15-fold fold increase that is indicated is not apparent from the western blot that is being presented. How were the bands quantified to calculate the p-P65/P65 ratio?
Response: Thanks for this comment. We explained the findings of the A549 experiment in more detail in the revised manuscript. “P65 NF-κB (P65) signal pathway is the classical pro-inflammatory signal pathway, which play a critical role in the innate immunity. And phosphorylated P65 (p-P65) is the active form of P65 NF-κB. The increasing of p-P65/ P65 means that this pro-inflammatory signal pathway is activated. In the present study, AT149 transfection could increase RIG-I expression and the p-P65/ P65, suggesting that AT149 5’PPP dsRNA induce RIG-I increasing to activate the following P65 NF-κB pro-inflammatory signal pathway.” (Line 170-175, Page 4).
In this study, we measured the gray values of p-P65 and P65 by Image J software and then the above two gray values were used for ratio. It means that p-P65/ P65 = the gray values of p-P65/ the gray values of P65. We added this description in the Materials and Methods. (Line 118-120, Page 3)
- Line 173 – The authors use the word ‘visibly’ several times. Since they perform statistical significance testing, the word ‘significantly’ would be better. Check throughout manuscript.
Response: Thanks for this comment. We have changed the ‘visibly’ into ‘significantly’ in the whole manuscript.
- Line 212 – Have the authors also measured any Th2 responses? One of the concerns with the COVID-19 vaccines has been the risk of vaccine-enhanced disease through a Th2 >> Th1 immune response.
Response: Thanks for this helpful comment. In this study, we have only measured the Th1 response. Indeed, the published studies show that one of the concerns with the COVID-19 vaccines has been the risk of vaccine-enhanced disease through a Th2 >> Th1 immune response. We will perform to measure the Th2 response in future studies. (Line 240-243, Page 6). And we added it in the revised manuscript. Thanks for this comment again.
- Figure 2A depicts a high-level overview of the study design, not a ‘detailed experimental procedure’.
Response: Thanks for this comment. We revised it in the manuscript. “The high-level overview of the animal study design” (Line 136-137, Page 3 and Line 256, Page 7).
- References – check references for consistency. For example, some references contain full journal names and full first names instead of initials.
Response: Thanks for this comment. We revised it in the manuscript.
Suggestions for changes to the text
- Lines 42-45 – The authors mention that ‘The accepted approach to develop a broad-spectrum recombinant protein vaccine is to design broad-spectrum antigens and combine them with more effective novel adjuvants that can generate higher immunogenicity and broader cross-protective immunity’. However, to increase the breadth of vaccines, most often the number of antigens is increased (e.g., Prevnar and Gardasil). Suggestion to amend text to: ‘One approach to develop a broad-spectrum recombinant protein vaccine is to design broad-spectrum antigens and combine them with adjuvants resulting in higher immunogenicity and broader cross-protective immunity [4].’
Response: Thanks for this helpful comment. We revised it in the manuscript. And we used “design broad-spectrum antigens and combine them with novel adjuvants” (Line 50-53, Page 2). Because novel vaccine adjuvant is very important for recombinant protein vaccine.
- Lines 45-46 - CpG or poly(I:C) have been used in clinical trials for many years, so ‘Recently,’ should be deleted. Furthermore, it is not clear why these two adjuvants are mentioned.
Response: Thanks for this comment. In the revised manuscript, we deleted “Recently”. Besides, we added the description to explain the reason of these two adjuvants mentioned. “Ligands of pattern recognition receptors are widely studied as vaccine adjuvants, such as CpG and Poly (I:C) in clinical trial. Both CpG and Poly (I:C) are nucleic acid adjuvants and they are used in the development of COVID-19 recombinant protein vaccines.” (Line 53-56, Page 2). “Poly (I:C) and CpG are the typical nucleic acid adjuvants, which are widely studied and presented good adjuvant effects now. In clinical trials, Al + CpG7909 has been used as a compound adjuvant in SARS-CoV-2 recombinant protein vaccines” (Line 200-203, Page 5).
- Lines 47-48 - Suggestion to change to: ‘The retinoic acid-inducible gene-I (RIG-I) receptor is widely expressed in immune and non-immune cells and is activated by short 5′triphosphate double strain RNA (dsRNA). This will lead to activation of the innate immune system to produce pro-inflammatory cytokines and induce B cell activation, suggesting that RIG-I agonists have potential as vaccine adjuvants [8, 9].
Response: Thanks for this helpful comment. We changed it in the revised manuscript.
- Line 51 – Please, clarify to what GUAU was compared or delete ‘more’ from the sentence.
Response: Thanks for this helpful comment. We delete “more” in the revised manuscript.
- Line 52 – Check instruction for authors on referencing unpublished work.
Response: Thanks for this comment. The unpublished work was performed by ourselves. And we noted it in the revised manuscript.
- Lines 63-65 – Delete sentences, as they are already mentioned in the introduction.
Response: Thanks for this comment. We deleted it in the revised manuscript.
- Lines 66-69 – Amend to: ‘dsRNA was produced via in vitro transcription (IVT). The sequence of the dsRNA, called AT149, used in this study was as follows:
5’GAUUAGUAUGUAUUAGUAUGUAUGUAUAUUGUAUGUAUGUAUGUAUUAGUAUGUAUGUAUUAGUAUGUAUUAACGGUUAGUGUGUGUUAGUGUGUGUGUGUUAGUGUGUGUGUGUGUGUAAUGUGUGUGUGUGUUAGUGUGUGUUAAUC-3’
Response: Thanks for this comment. We changed it in the revised manuscript.
- The authors mention that AT149 ‘was designed based on our previous study, which activated the RIG-I signaling pathway substantially [10].’ However, it is not clear from reference 10 why a GUAU-rich sequence was chosen.
Response: We feel apologized it due to our neglect. Reference 10 was not clear the reason of GUAU-rich sequence. We deleted this reference in the revised manuscript. Chose GUAU-rich sequence was based on our unpublished observation. We have stated it in the manuscript. (Line 61-62 and Line 73-74, Page 2).
- Lines 84-85 – Please, provide exact names of HEK293 reporter cell lines from Invivogen and check the description of the reporter cell lines. Note that the Invivogen website mentions that in HEK-Lucia cell lines, Luciferase expression is under the control of the interferon (IFN)-inducible promoter which is comprised of the IFN-stimulated genes (ISG) 54 promoter (see https://www.invivogen.com/hek-lucia-null).
Response: Thanks for this suggestion. We provided exact names in the revised manuscript. “HEK-Lucia™ Null cells with RIG-I overexpression (OE), RIG-I knockout (KO), and RIG-I Null purchased from InvivoGene (San Diego, CA, USA) are HEK293-derived cells that stably express the Lucia luciferase secreted reporter gene. This reporter gene is under the control of the interferon (IFN)-inducible promoter which is comprised of the IFN-stimulated genes (ISG) 54 promoter enhanced by multimeric IFN-stimulated response elements (ISRE). The detail information was recorded in https://www.invivogen.com/hek-lucia-null.” (Line 97-103, Page 3).
- Line 108 – amend to ‘AT149 cells were was encapsulated….’
Response: We feel apologized it due to our neglect. We changed it in the revised manuscript. AT149 was encapsulated in lipid nanoparticles (Line 128-129, Page 3).
- Lines 114-125 – The rationale for the experiment should be mentioned in the Results section. Proposed change: ‘For study 1, 30 Balb/c mice (6–8 weeks old, 18–20 g) were randomly and equally assigned to six groups as detailed in Table 1. For Study 2, 60 BALB/c mice (6–8 weeks, 18–20 g) were randomly assigned to ten groups as detailed in Table 2. An overview of the experimental procedure for both studies is shown in Figure 2A.’
Response: Thanks for this comment. We changed it in revised the manuscript.
- Lines 158-160 – Delete ‘and the abundant GUAU sequence to investigate whether it could improve the immunogenicity and the broad spectrum of recombinant protein vaccines.’
Response: Thanks for this comment. We deleted it in the revised manuscript.
- Line 173 - Delete ’could’.
Response: Thanks for this comment. We deleted it in the revised manuscript.
- Line 175 – Suggestion: ‘Subsequently, AT149 adjuvanted Delta-Omicron BA1 chimeric RBD-dimer recombinant protein (D-O RBD) was used to immunize mice two times with one-fifth of the human dosage (Figure 2 A).’
Response: Thanks for this comment. We changed it in the revised manuscript.
- Lines 183-186 – Suggestion: ‘In the present study, Al and AT149 were combined. Results showed higher Delta and Omicron BA1 specific IgG titers for D-O RBD with Al + AT149 (5–20 μg) compared to D-O RBD + Al + CpG7909/ Poly (I:C) groups (Figure 2B).’
Response: Thanks for this comment. We changed it in the revised manuscript.
- Line 190: ‘13- to 59-fold’ and ‘16- to 38-fold’.
Response: Thanks for this comment. We changed it in the revised manuscript.
- Lines 191-194 – Suggestion: ‘Surprisingly, cross-neutralizing antibody titers against Omicron BA5 and BF7 were also increased by 7- to 21-fold in the D-O RBD + AT149 (5–20 μg) groups and were 21- to 31- fold higher than in the D-O RBD + Al group (Figure 2C).’
Response: Thanks for this comment. We changed it in revised the manuscript.
- Suggestion to mover comments on transmissibility and immune evasion of variants from lines 191-192 and 195-197 to line 209: ‘….and provide more effective cross-neutralization against recently emerged, highly transmissible variants Omicron BA5, BF7, BQ1.1, and XBB, which have shown increased immune evasion.’
Response: Thanks for this comment. In this part, we first investigated the efficacy of AT149 as adjuvant and then the efficacy of AT149 + Al as compound adjuvant. Thus, we first describe the results of D-O RBD + AT149 and then the results of D-O RBD + AT149 + Al. Thank you again.
- Line 211 – what does ‘resisting SARS-CoV-2 invasion’ mean?
Response: Thanks for this comment. We feel apologized it due to our misstatement. We want to show that cytotoxic T lymphocytes could kill viruses. Thus, we deleted ‘resisting SARS-CoV-2 invasion’ in the revised manuscript.
- Figure 2 – A larger figure is needed to be able to read some of the text within the figure. Upon printing, the fold increase that is mentioned in most panels is hardly readable.
Response: Thanks for this helpful comment. We have enlarged the word size in the revised manuscript.
- Line 226 – see earlier comment on ‘high-level overview’ vs ‘detailed experimental procedure’.
Response: Thanks for this comment. We changed it in the revised manuscript.
- Line 240 – ‘…broaden the spectrum…’
Response: Thanks for this comment. We changed it in the revised manuscript.
We really appreciate for the earnest work of Editors and Reviewers, and hope that the corrections will meet with approval. Once again, thank you very much for your comments and suggestions. We look forward to your information about my revised manuscript and thank you for your good comments.
Best regards!
Yours sincerely,
Dr. Zhenglun Liang
National Institutes for Food and Drug Control. Huatuo Road No.31, Daxing District, Beijing, China. 102629
Email: [email protected]

Reviewer 4 Report
I do not have a comments and suggestions for Authors. It is not easy to prepare a very good vaccine not only therapeuticly but mainly preventively.
Author Response
Responses to reviewer 4’ comments for the manuscript (NO. viruses-2366845)
Dear Editor/ reviewer,
We are very grateful to editor and reviewer for the interest in our work and those valuable advices and comments. We have adopted the suggestion and revised them accordingly. The revised parts were marked in green in the revised manuscript.
Reviewer Comments
(1)I do not have a comments and suggestions for Authors. It is not easy to prepare a very good vaccine not only therapeutical but mainly preventively.
Response: Thank you. We have revised this paper carefully to improve the quality.
We really appreciate for the earnest work of Editors and Reviewers, and hope that the corrections will meet with approval. Once again, thank you very much for your comments and suggestions. We look forward to your information about my revised manuscript and thank you for your good comments.
Best regards!
Yours sincerely,
Dr. Zhenglun Liang
National Institutes for Food and Drug Control. Huatuo Road No.31, Daxing District, Beijing, China. 102629
Email: [email protected]

Round 2
Reviewer 3 Report
Although the authors explained the method to determine the p-P65:P65 ratios, the outcome is confusing. In principle the amount of p-P65 cannot be higher than the total amount of P65, so the ratio cannot be higher than 1.
Instead the authors should refer to 'fold-increase in p-P65' in the text and the y-axis of the left-hand figure in Fig.2E should be amended accordingly.
English can still be improved.
Some examples:
Line 95 - The sample was tested according to the instruction manual of the instrument.
Lines 118-120: Gray values for phosphorylated P65 (p-P65) and total P65 were measured using Image J software and used to calculate the fold-increase in p-P65.
Line 167: The capillary electrophoresis results showed that the prepared AT149 eluted as a single peak with a purity of 86.7%.
Author Response
Responses to reviewer 3’ comments for the manuscript (NO. viruses-2366845)
Dear Editor/ reviewer,
We are very grateful to editor and reviewer for the interest in our work and those valuable advices and comments. We have adopted the suggestion and revised them accordingly. In the revised manuscript, the changed parts were marked in blue in this revision.
Comments and Suggestions for Authors
- Although the authors explained the method to determine the p-P65:P65 ratios, the outcome is confusing. In principle the amount of p-P65 cannot be higher than the total amount of P65, so the ratio cannot be higher than 1.
Response: Thanks for this helpful comment. As you said, the amount of p-P65 cannot be higher than the total amount of P65 in cells. But, the membranes of p-P65 and P65 needed to be exposed twice in the CCD instrument. Please, attention. The results from the twice different exposures cannot be analyzed and compared in WB experiment. Only, samples on the same membranes can be analyzed and compared under the same exposure condition. In this study, the ratio of p-P65/ P65 was higher than 1. Nonetheless, it does not mean that p-P65 was higher than the total amount of P65. β-actin is the reference protein in cells. In order to eliminate possible misunderstandings to readers, we used p-P65/ β-actin to show the increase or decrease of the expression of p-P65. Thus, we changed it to 'fold-increase in p-P65/ β-actin ' in Figure 1 E.
- Instead, the authors should refer to 'fold-increase in p-P65' in the text and the y-axis of the left-hand figure in Fig.1E should be amended accordingly.
Response: Thanks for this valuable comment. In the revised manuscript, we changed it to 'fold-increase in p-P65/ β-actin ' in the text and the y-axis of the left-hand figure in Figure 1 E. (Line 118-120, Page 3).
Comments on the Quality of English Language
Thanks for your careful job for our manuscript. The whole manuscript was edited again under the help of a native English speaker.
- Line 95 - The sample was tested according to the instruction manual of the instrument.
Response: Thanks for this helpful comment. We changed it in the revised manuscript.
- Lines 118-120: Gray values for phosphorylated P65 (p-P65) and total P65 were measured using Image J software and used to calculate the fold-increase in p-P65.
Response: Thanks for this comment. We changed it in the revised manuscript. Thanks again.
- Line 167: The capillary electrophoresis results showed that the prepared AT149 eluted as a single peak with a purity of 86.7%.
Response: Thanks for this comment. We changed it in the revised manuscript.
We really appreciate for the earnest work of Editors and Reviewers, and hope that the corrections will meet with approval. Once again, thank you very much for your comments and suggestions. We look forward to your information about my revised manuscript and thank you for your good comments.
Best regards!
Yours sincerely,
Dr. Zhenglun Liang
National Institutes for Food and Drug Control. Huatuo Road No.31, Daxing District, Beijing, China. 102629
Email: [email protected]
